# Developing a Tuned Three-Layer Perceptron Fed with Trained Deep Convolutional Neural Networks for Cervical Cancer Diagnosis

**DOI:** 10.3390/diagnostics13040686

**Published:** 2023-02-12

**Authors:** Shervan Fekri-Ershad, Marwa Fadhil Alsaffar

**Affiliations:** 1Faculty of Computer Engineering, Najafabad Branch, Islamic Azad University, Najafabad 8514143131, Iran; 2Big Data Research Center, Najafabad Branch, Islamic Azad University, Najafabad 8514143131, Iran; 3Department of Medical Laboratory Techniques, Al-Mustaqbal University College, Hillah 51001, Babylon, Iraq

**Keywords:** cervical cancer diagnosis, Pap smear image classification, deep learning, multi-layer perceptron neural network, feature extraction

## Abstract

Cervical cancer is one of the most common types of cancer among women, which has higher death-rate than many other cancer types. The most common way to diagnose cervical cancer is to analyze images of cervical cells, which is performed using Pap smear imaging test. Early and accurate diagnosis can save the lives of many patients and increase the chance of success of treatment methods. Until now, various methods have been proposed to diagnose cervical cancer based on the analysis of Pap smear images. Most of the existing methods can be divided into two groups of methods based on deep learning techniques or machine learning algorithms. In this study, a combination method is presented, whose overall structure is based on a machine learning strategy, where the feature extraction stage is completely separate from the classification stage. However, in the feature extraction stage, deep networks are used. In this paper, a multi-layer perceptron (MLP) neural network fed with deep features is presented. The number of hidden layer neurons is tuned based on four innovative ideas. Additionally, ResNet-34, ResNet-50 and VGG-19 deep networks have been used to feed MLP. In the presented method, the layers related to the classification phase are removed in these two CNN networks, and the outputs feed the MLP after passing through a flatten layer. In order to improve performance, both CNNs are trained on related images using the Adam optimizer. The proposed method has been evaluated on the Herlev benchmark database and has provided 99.23 percent accuracy for the two-classes case and 97.65 percent accuracy for the 7-classes case. The results have shown that the presented method has provided higher accuracy than the baseline networks and many existing methods.

## 1. Introduction

Cervical cancer is one of the most common types of cancer among women, which has higher death-rate than many other cancer types. [1]. Medical studies have shown that early detection of this type of cancer can greatly increase the effectiveness of treatment solutions such as chemotherapy and radiation therapy, and reduce the possibility of the patient’s death [1]. When cervical cancer is detected at an early stage, the 5-year survival rate for people with invasive cervical cancer is 92%. However, just about 44% of women with cervical cancer are diagnosed at the early stages. If cervical cancer has spread to surrounding tissues or organs, the 5-year survival rate decreases to 58%. In most cases, cervical cancer occurrs due to the abnormal growth of cells, which may spread to other parts of the body. The Pap smear test is an efficient tool to detect potentially cancerous processes by screening the cervix [2]. In this test, different cells from the whole cervix are collected as images. Next, they are analyzed to find dysplasia or precancerous variations. Cells in the cervix have two main parts called the nucleus and cytoplasm. Usually, the nucleus and cytoplasm are stable structures in the body of a healthy person regarding size, appearance, texture, color, etc. Therefore, any change in these characteristics can be an early sign of cervical cancer. For example, in healthy cells, the size of the cytoplasm is larger than the nucleus by nearly a same constant ratio. However, in cancer cells, the size of the nucleus often increases and the ratio of cytoplasm to nucleus decreases. Some examples of normal and abnormal cells in Pap smear images from the Herlev dataset [3] is shown in Figure 1. The visual differences can be seen in terms of the nucleus-to-cytoplasm ratio, texture, shape, etc.

Today, the Pap smear test is widely used to diagnose cervical cancer. During cyto-technician analysis the cells are removed from the cervix and are placed under a microscope. The Pap smear test can be effective when analyzed correctly. So far, various methods have been proposed for the diagnosis of cervical cancer based on the analysis of Pap smear images automatically. The aim of all artificial-intelligence-based methods is the accurate classification of Pap smear images [4]. So far, many articles have been presented in this field. In some areas of the world where it is difficult to access a specialist doctor, the automatic diagnosis system can help the early diagnosis of patients and connect patients to the nearest specialist doctor and surgeon. Additionally, the costs for patients and insurance companies will decrease. Automatic detection systems can be replicated in all regions around the world thanks to simple software. Most of the methods that have been presented so far can be classified into two categories: methods based on machine learning and methods based on deep learning [5]. Machine-learning-based methods usually include two stages of feature extraction and classification. In this group of methods, firstly, features are extracted from Pap smear images, and then, training process is performed. Finally, in the classification phase, the trained system predicts the label of the test image.

In deep-learning-based methods, the process of feature extraction is also conducted, but this process takes place in the learning phase and is not implemented as a separate phase. In the structures of deep networks, popular layers, such as fully connected or softmax, are usually used in the last layers to classify the test image.

Deep-learning-based methods usually provide higher accuracy than methods based on machine learning, but their computational complexity and runtime are higher and their speed is lower. Cervical cancer diagnosis is an important issue for humans and usually there is no need for online diagnosis in healthcare systems. In most cases, the patient is willing to spend a reasonable amount of time to receive a more detailed and accurate analysis. Deep-learning-based methods cannot be ignored due to their high accuracy in classification. Therefore, in this article, a multi-layer perceptron (MLP) neural network is designed, which feeds from two deep networks, ResNet34/ResNet-50 and VGG-19. In this regard, first the layers related to the classification phase have been removed from these two deep networks and replaced with a flatten layer to build the input material of designed MLP. In this article, deep neural networks are used just for feature extraction. Therefore, some of the final layers in these networks, which are related to the classification activity, have been removed. Hence, the runtime of the proposed method is somewhat lower than pure deep neural networks. The runtime required to classify one input image in the presented method and two basic deep networks has been evaluated in the results section. The results show that removing the fully connected or softmax layers, reduced the total runtime. The efficiency of the proposed method has been evaluated on the benchmark database called Herlev [3]. The Herlev database is classified in two ways. As two classes (healthy and diseased) and seven classes (three types of normal cells and four types of abnormal cells). Therefore, the proposed method has been evaluated in both cases and compared with the methods used in this field. The comparative results demonstrate the main contribution of this paper.

### 1.1. Main Contribution and Novelty

The main goal of this article is to present a method for diagnosing cervical cancer based on Pap smear image analysis. Most of the methods that have been presented so far can be classified into two categories, based on machine learning or deep-learning-based. Feature engineering is not used in deep-learning-based methods, and feature extraction and classification steps are not separate. Therefore, the main contribution of this paper is to present a method based on machine learning strategy for cervical cancer diagnosis, which uses deep features. Therefore, in this article, the feature extraction stage is performed separately based on deep networks, and a multi-layer perceptron network is used in the classification stage. Different deep networks (such as ResNet, VGG, GoogleNet, etc.) can be used to extract deep features. This paper presents a general approach for using the potential of deep features in a machine-learning-based structure for cervical cancer diagnosis. In the classification stage, a MLP is used, with some simple ideas to tune the number of hidden neurons to increase the detection accuracy.

As novelty, our researches indicate that so far no paper has used the deep features extracted from two deep networks, VGG-19 and ResNet-34, together with the tuned three-layer perceptron neural network to diagnose cervical cancer types. In comparison with state-of-the-art methods in this scope, our proposed approach introduces two main novelties:In this article, the feature map of the non-terminal layers of VGG-19 and ResNet-34/ResNet-50 are used to diagnose cervical cancer in a machine learning strategy format;For the first time, a tuned multi-layer perceptron neural network, in terms of hidden neurons, fed with deep features is used to diagnose cervical cancer.

### 1.2. Paper Organization

The rest of this article is organized as follows: In Section 2, related works in this field are examined. In Section 3, firstly, the ResNet34 and VGG-19 networks are explained, and then the proposed method for diagnosing cervical cancer in Pap smear images is presented. In Section 4, the database and evaluation criteria are introduced, and then the performance of the proposed method is evaluated. At the end of this section, the presented method is compared with efficient methods in this scope. The conclusion is given in Section 5.

## 2. Related Works

Most of the methods that have been presented so far for the classification of Pap smear images have two main stages of learning and classification. In some methods, side steps, such as image preprocessing or parameter optimization, are also provided. Feature extraction is performed before the learning stage or simultaneously with it. Therefore, in the continuation of this section, some of the most efficient methods in this field are studied from the two perspectives of feature extraction and classification process.

Arya et al. [6] use a set of texture feature for Pap smear image classification. The proposed approach by Arya et al. [6] consists of two phases, feature extraction and classification. First of all, popular handcrafted texture features, such as gray-level co-occurrence matrixes (GLCM), local binary patterns (LBP), Law’s and discrete wavelet transform (DWT), are extracted from input single-cell images. Next, two different classifiers, such as the linear support vector machine (SVM) and artificial neural network (ANN), are performed for classification phases. The reported results show that ANN in joint of combination of above-mentioned extracted features provide maximum accuracy when compared with SVM [6].

Fekri-Ershad [7] proposed an approach for Pap smear classification based on the combination of different statistical and computational features. Time series features in joint of textural features, such as mean, entropy and correlation, are used in the feature extraction phase. Additionally, global significant value is added as an innovative feature to improve the final accuracy. The performance of the proposed approach in [7] is evaluated based on different linear classifiers, such as the Bayesian network, naïve Bayes and KNN in the Herlev dataset. Finally, the classification accuracy of about 88.47 percent is provided for the 2-classes classification case [7]. A set of statistical and numerical features are suggested in [7], which can be used in some other related computer vision applications, such as the characterization of bacteriophages from sewage water [8] or silver nanoparticles detection [9].

Bora et al. [10] used ALexNet as an efficient CNN for Pap smear classification. In [10], firstly, Alexnet is performed to extract deep features. Next, to decrease the dimensions, the unsupervised feature election technique is used. Finally, different classifiers, such as LSSVM and softmax regression are performed to classify Pap smear test image. As reported in [10], two layers of LSSVM provide the highest accuracy in the Herlev dataset, about 94.61 percent, in comparison with other classifiers. The AlexNet-based architecture of the CNN that is proposed in [10] is shown in Figure 2. As can be seen, the proposed CNN should be fully performed for feature extraction. So, feature selection just decreases the input size of classifiers and it cannot reduce the total computational complexity or runtime of this approach.

Zhang et al. [11] designed an innovative deep network called ConvNet-T for Pap smear image classification. ConvNet-T is pre-trained using a dataset consisting of natural images. Subsequently, ConvNet-T is fine-tuned in a dataset consisting of re-sampled image windows from cervix cells. Additionally, the aggregation technique is used to average the prediction scores of different image windows. The ConvNet-T provides classification accuracy about 98.3 percent in the Herlev dataset. The total structure of the proposed approach in [11] is shown in Figure 3. As can be seen, the proposed CNN should be tuned twice. Additionally, data aggregation and transfer learning are performed in the classification process. So, the computational complexity of this approach is too high. Additionally, it is too time-consuming and expensive to collect a pure dataset of cervix cell samples [11].

Bora et al. [12] used the combination of color, texture, and shape features to analyze Pap smear image. In the preprocessing step of [12], three filters were performed to remove noise, RBC, and inflammatory cells. Next, different texture, color and shape features were extracted from image. Finally, three different classifiers were employed to predict the label of the test image. In this step, MLP, RF, and LSSVM were used. Finally, weighted majority voting is performed between predicted labels and the final label is reported. Voting is performed between three classifiers in the method [12], so it cannot be used for more than a binary classification problem. Additionally, a straightforward approach is not described in [12] to tune weights. In a recent study, a novel cervical cell classification approach is proposed by Shi et al. [13], using a graph convolutional neural network. First of all, CNN is used to extract the deep features of all the cervical cell samples. Next, the relationships of images can be preliminarily revealed through the clustering. To analyze the correlations that existed among the above clusters, a graph structure was built [13]. Yu et al. [14], proposed four different models for Pap smear image classification. All of the four models were constructed based on different possible joints of CNN with inspection technique and/or spatial pyramid pooling for cervical cancer detection in Pap smear images [14].

In a recent study, Fekri-Ershad and Ramakrishnan [15] used texture features in joint of multi-layer neural network for cervical cancer diagnosis. Modified uniform local ternary patterns (MU-LTP) are used in [15] to extract discriminative texture features. Next, a multi-layer neural network is designed for the classification phase. A genetic algorithm is performed to tune the number of hidden neurons in the proposed neural network. A classification accuracy of about 98.90 percent is reported in [15]. In the genetic algorithm, multi epochs should be performed to select the best chromosomes. So, the computational complexity of the proposed approach in [15] is higher than some other methods in this scope. Hosseinabadi et al. [16] used the combination of the gray-level co-occurrence matrix, local binary patterns, and rotational histogram to extract features. Additionally, different supervised classifiers are used for the classification phase, which support the vector machine (SVM) that provides maximum accuracy [16]. Shanthi et al. [17] proposed a literature survey on Pap smear classification methods such as machine-learning-based and deep-learning-based methods. Fang et al. [18] constructed a deep convolutional neural network with feature representations learned via multiple kernels in different sizes, which is called DeepCELL. Firstly, three different basic modules of DeepCELL were designed to extract features using multiple kernels. Next, the classification was performed using a softmax layer with two or seven neurons [18].

## 3. Proposed Cervical Cancer Diagnosis Method

As described above, Pap smear classification is used in this study to diagnose cervical cancer. Hence, our proposed approach consists of two main steps, train and test. Learning process of multi-layer neural network and feature extraction process are performed simultaneously in the train step. A prediction of the label of the test image is performed in the test step. The main block diagram of our proposed method is shown in Figure 4. All of the boxes in the train and test stages are describe in detail in the following sub-sections.

### 3.1. ResNet34

In some deep neural networks, with the network depth increasing in the converging process, the accuracy becomes saturated and then degrades rapidly [19,20]. To solve this problem, residual neural network, ResNet as acronym, was presented by He et al. [21] for the first time and won the ImageNet competition in 2015. Instead of hoping that every few stacked layers directly fit a desired underlying mapping, they explicitly let these layers fit a residual mapping [21]. Some advantages of ResNet are as follows:ResNets are easy to optimize compared with some previous networks such as VGG or AlexNet;ResNets can easily gain accuracy from greatly increased depth;ResNet provides results with higher performance than some previous networks in image classification cases.

ResNet-34 is a variant of ResNet, which is widely used in computer vision applications such as defect detection [22], medical diagnosis [23], image segmentation [24], handwritten identification [25], emotion recognition [26], etc. ResNet-34 consists of 34 convolutional layers, one max pooling layer in the size of 3 × 3, one average pooling layer, and finally, a fully connected layer. Resnet34 is a CNN-based network, which is pre-trained on the ImageNet dataset in basic architecture. The basic ResNet-34 network consist of 63.5 million parameters, which is lower than some popular deep networks, such as VGG-19 and DenseNet. Nonlinearity (ReLU) as an activation function and batch normalization (BN) are applied to the back of all convolution layers in the ResNet34. The main architecture of VGG-19 layers is shown in Figure 5.

### 3.2. VGG-19

VGG-19 is a convolutional neural network (CNN) that was presented for first time by the Visual Geometry Group of Oxford University in 2014 [27]. On the first experiment, VGG-19 obtained high classification accuracy on the ImageNet dataset [28]. VGG19 is a version of the basic VGG model, which consists of the 19 layers. A total of 16 convolution layers, three fully connected layers, five max pooling layers and one softmax layer are joined sequentially in these 16 layers. VGG19 has 19.6 billion FLOPs and 138 million parameters. In recent studies, VGG-19 has been used in medical image analysis [29], image classification [30], agriculture product inspection [31], etc. The concept of the VGG19 model is the same as the VGG16; VGG19 has three more convolutional layers than VGG16. VGG19 provides discriminative features and a good understanding of basic image properties such as shape, color, and texture. The main architecture of the VGG-19 layers is shown in Figure 6.

### 3.3. Proposed Multi-Layer Neural Network for Cervical Cancer Diagnosis

As mentioned above, the cytoplasm and nucleus of abnormal cells in Pap smear images are so different compared to normal cells in terms of texture, size of nucleus, content, and contrast. So, it is so important to describe these types of cells with features that describe these properties together. The VGG-19 network is first introduced as a way of extracting the content and style features of the images. The pre-trained VGG-19 network is evaluated on the CIFAR-10 dataset and achieved 91.8 percent accuracy. CIFAR is a huge dataset of natural images in different types. So, extracted features using VGG-19 in the learning process are useful to describe Pap smear images.

ResNet34 is quite a shallow network, while VGG19 is considered a deeper network. ResNet34 provides accurate performance in many medical pattern classification problems. Due to the connection of the initial input of the residual block to the output of the last layer in this block, the ResNet34 network extracts deep features in the presence of low-level features. In the analysis of Pap smear images, some properties of abnormal cases, such as the enlargement of the ratio of nucleus to cytoplasm, can also be seen visually. Extracting deep features should not lead to ignoring these low-level properties. Therefore, in this article, a model for classifying Pap smear images based on an innovative multi-layer perceptron fed with VGG-19 and ResNet-34 is introduced. The main architecture of the proposed method is shown in Figure 7.

The ResNet34 network uses only one fully connected layer for the classification phase. Additionally, the VGG-19 network uses three fully connected layers and one softmax layer for classification. The classification architecture in these two networks is not the same. Therefore, in order to be able to use the advantages of these two CNNs together, in the proposed model, the classification’s layers are removed from the network structure and the feature map is extracted from the last pooling layer (max. pooling in Vgg-19 and average pooling in ResNet34) and is placed into a flatten layer. After converting the feature map to the feature vector in flatten layers, the two extracted feature vectors connect to each other (concatenating layer) and enter as input to a multi-layer perceptron neural (MLP) network. Therefore, the proposed MLP network feeds on these extracted deep features. The number of neurons in the last layer of the MLP network is equal to the number of classes in the database. Dataset details and the number of classes are described in detail in Section 4.1. As described above, the VGG-19 and ResNet-34 are used in our proposed approach as feature extractors to feed the tuned MLP. In order to improve their performance, VGG-19 and ResNet-34 have been trained using a collection of 80 Pap smear images. Hyper-parameter optimization is described in Section 3.3.2.

As discussed in the introduction, the proposed framework is a general case that uses deep features to feed the MLP. So, it is possible to use different deep neural networks in the feature extraction phase. There are more ResNet versions that have the similarity of using residual blocks. For example, the ResNet-50 structure includes a 50-layer convolutional neural network (48 convolutional layers, one max. pooling layer, and one average pooling layer). ResNet-50 used a stack of three layers instead of the earlier two. Hence, each one of the two-layer blocks in Resnet-34 is replaced with a three-layer bottleneck block. We evaluated the performance of our proposed approach using ResNet34 + VGG19 and ResNet50 + VGG19 in the Results section to discuss generality.

#### 3.3.1. Tuned MLP

MLP is a network of connected simple *neurons* called *perceptrons*. Each perceptron computes one *output* from multiple real-valued *inputs* by forming a linear combination based on its input *weights* and then possibly putting the output through some nonlinear activation function (Equation (1)).
(1)$$Y=\varphi \left(\overset{n}{\underset{i=1}{\sum }}w_{i}x_{i}+b\right)$$
where *W* denotes the vector of weights and *x* is the vector of inputs. Additionally, *b* is the bias value and φ shows the activation function. In our proposed method, the size of the input vector can be calculated as follows (Equation (2)):n = SF_VGG-i_ + SF_ResNet-i_(2)
where, SF_VGG-i_ shows the number of flatted features of the ith layer of the VGG-19 and SF_ResNet-i_ shows the number of features after passing flatten layer of the ith layer of ResNet-34. Nowadays, especially in multilayer networks, the activation function is often chosen to be the logistic sigmoid 1/(1 + e^−x^). This function is used because it is mathematically convenient and is close to the linear near origin while saturating rather quickly when moving away from the origin. This allows MLP to model well both strongly and mildly nonlinear mappings.

Actually, one single perceptron is not useful because of its limited mapping ability. The perceptron is only able to represent an oriented ridge-like function. So, a multiplicity of perceptrons is suggested. A typical MLP consists of a set of source nodes forming the *input layer*, one or more *hidden layers* of computation nodes, and an *output layer* of nodes. The input signal propagates through the network layer by layer. The computations performed by such a feedforward network with a single hidden layer with nonlinear activation functions and a linear output layer can be calculated as follows (Equation (3)):(3)Y=Bφ(AX+a)+b
where X is a vector of inputs and Y a vector of outputs. A shows the matrix of weights of the first layer, a is the bias vector of the first layer. B and b are, respectively, the weight matrix and the bias vector of the second layer. The function φ denotes an elementwise nonlinearity.

The number of hidden layers and hidden neurons are variable parameters in the MLP, which may be effect the final performance of the proposed approach. In [32], the authors performed a genetic algorithm to tune these parameter values. In the genetic algorithm, the fitness of each chromosome should be computed once. The computational complexity of performing evolutionary algorithms in deep learning methods is too high. Four novel theories are proposed to select the number of hidden neurons of an MLP. The number of neurons in the hidden layer of the MLP network is an adjustable parameter that also plays a role in the final classification accuracy. Because in this article deep networks are used for feature extraction, usually the dimensions of the feature map in the pre-end layers in these networks are large. Logically, the number of neurons in the MLP network can be any integer value, so determining a logical limitation for this, which makes it applicable in all other problems, plays an important role in choosing the four proposed ideas. Therefore, we suggested four ideas based on following issues:(a)The number of features and the number of classes are the only parameters common to all classification problems. Therefore, these two parameters were used in all four proposed ideas;(b)In most classification problems, the number of features is usually more than the number of classes. In some databases, the number of classes is slightly more than the number of features. Therefore, limiting the number of neurons in the four proposed ideas based on these two points and their average was attempted.

These theories are as follows.
I.Number of classes;II.Number of attributes;III.Number of attributes + Number of classes;IV.(Number of attributes + Number of classes)/2.

The complexity of using evolutionary algorithms such as genetic techniques is too high in our case. So, these four theories are used in our experiments to select MLP parameters as reported in the next section. In most datasets, the number of features is higher than the number of classes. However, this does not occur in all cases. So, both of these statuses are considered in the proposed theories. Additionally, in most cases, the height of the hidden layer of MLP is suggested to be more than the input and output layer to improve learning. So, it is considered in the third and fourth theories.

#### 3.3.2. Hyper-Parameter Optimization

As described above, the VGG-19 and ResNet-34 are used in our proposed approach as feature extractors to feed the tuned MLP. So, both of them are trained on 100 Pap smear images using Adam optimizer. Focal loss function [33] with γ = 2 is used as the loss function. To decrease computational complexity, both of them are trained in 30 epochs. The initial learning rate is considered to be 10^−5^ for first 10 epochs. Then, it is decreases to 10^−6^ for next 20 epochs. Xavier (Glorot) initialization is used for weights initialization. Glorot is an advanced initialization scheme for CNNs. The number of biases initialized is 0 and the weights at each layer are initialized (Equation (4)).
(4)$$W_{i,j}\sim \;U\left[-\frac{1}{\sqrt{S}},\;\frac{1}{\sqrt{S}}\right]$$
where U is a uniform distribution, and “S” shows the size of previous layer. Select the best loss function, which is a hyper-parameter depending on the problem we are facing. Multi-class cross-entropy loss seems a wise choice in many studies. However, according to the amount of distribution of cells in the body of each human and according to the database samples, we are facing an imbalanced classification problem. Focal loss is another choice whose properties we can leverage to enhance the performance of our model. This loss function tries to generate the class-weighting system in order to balance the samples in each batch size of data (Equation (5))
FL(p_t_) = −α_t_(1 − p_t_)^γ^ log(p_t_)(5)
where *p_t_* is a function of the true labels. Focal loss can be interpreted as a binary cross-entropy function multiplied by a modulating factor (1 − *p_t_*)*^γ^*, which reduces the contribution of easy-to-classify samples. The weighting factor α_t_ balances the modulating factor. There are several approaches for incorporating focal loss in a multi-class classifier. The one-versus-the-rest (OvR) technique, in which a binary classifier is trained for each class *C,* is used in this paper. The data from class *C* are treated as positive, and all other data as negative.

As described above, training our deep networks in 30 epochs provides maximum accuracy. In order to choose the best number of learning rates and epochs, our proposed method is evaluated in terms of different possible values. The results are reported in Table 1.

## 4. Experimental Results

### 4.1. Dataset

In order to evaluate the performance, a benchmark dataset in this scope called Herlev is used. The Herlev samples are classified in two types [3]. In the first type, all of the samples are classified in two classes, normal and abnormal. In this respect, 242 image samples are labeled as normal and 675 samples as abnormal. In the second type, sample mages are classified in seven classes as normal (superficial, intermediate, and columnar) and abnormal (dysplastic, moderate, severe, and carcinoma). All of the Herlev samples are saved in a BMP format and a 24 imaging bit depth. The image size and scale of the samples are not same, which make it harder to classify. Some examples of the Herlev dataset are shown in Figure 8.

The SIPaKMeD dataset includes 4049 images of cervix single cells. All of the images are collected with an optical microscope and one camera. SIPaKMeD are grouped into five classes, including dyskeratotic, koilocytotic, metaplastic, parabasal, and superficial/intermediate. The dyskeratotic and koilocytotic classes are known as abnormal cells, and the metaplastic class is benign cells. The remaining classes are considered normal cells [34].

### 4.2. Performance Evaluation Metrics

The main aim of this paper is to propose an accurate cervical cancer diagnosis system. In this respect, Pap smear classification is performed. So, this problem is defined as a pattern classification case. Classification accuracy is the main criterion that is used to evaluate the performance of this type of cases. As mentioned in Section 4.1, the Herlev dataset samples can be classified in two or seven classes. So, the classification accuracy can be calculated based on following equation for two class classifications.
(6)$${\mathrm{Accuracy}}_{2-\mathrm{classes}}=\left(\;\frac{\mathrm{TP}+\mathrm{TN}}{\mathrm{TP}+\mathrm{FP}+\mathrm{TN}+\mathrm{FN}}\;\right)\times 100$$

Additionally, accuracy in a multi-class classification problem can be calculated as follows:(7)$${\mathrm{Accuracy}}_{7-\mathrm{classes}}=\frac{\mathrm{correctly}\;\mathrm{classified}\;\mathrm{samples}\;}{\mathrm{total}\;\mathrm{samples}}\times 100$$

Cervical cancer diagnosis is a binary classification problem, where the risk of misdiagnosis of two classes is not the same. In other words, the risk of misdiagnosing an infected person as a healthy person is much higher than misdiagnosing a healthy person as a person with cervical cancer. In this respect, the performance of the proposed approach is evaluated in terms of precision and recall more than accuracy. These performance evaluation metrics can be calculated using Equations (8) and (9).
(8)$$\mathrm{Precision}=\left(\;\frac{\mathrm{TP}}{\mathrm{TP}+\mathrm{FP}}\;\right)\times 100$$
(9)$$\mathrm{Recall}=\left(\;\frac{\mathrm{TP}}{\mathrm{TP}+\mathrm{FN}}\;\right)\times 100$$

### 4.3. Performance Evaluation of Proposed Approach in Terms of Hidden Neurons

As mentioned above, the MLP is tuned in terms of number of hidden neurons based on four novel theories. The performance of our proposed model is evaluated based on four different theories to select MLP parameters in terms of accuracy. The experimental results of the proposed approach are reported in Table 2. As can be seen in Table 2, the fourth theory provides the highest accuracy in comparison with the other three theories and some user-selected values. The confusion matrix of our proposed method is based on the fourth theory on the Herlev dataset [3] in the two-class problem, which is reported in Table 3.

### 4.4. Performance Evaluation of the Proposed Approach in Terms of Different Classical Classifiers

As described in related sections, deep networks such as ResNet-34 and VGG-19 are used in this paper just to extract features. As mentioned above, a tuned three-layer MLP is used for classification. Feature engineering was not performed in this paper, and only the output of the pre-final layer of two deep networks was used for feature extraction. Therefore, for the classification phase, a simple perceptron neural network was used and an attempt was made to increase the final classification accuracy by adjusting the number of neurons in the inner layer. However, it is possible to perform different machine-learning-based classifiers to classify Pap smear images. The performance of the proposed approach is evaluated based on different classical classifiers on two datasets (Herlev and SIPKAMED) in terms of accuracy (Table 4 and Table 5).

Deep features are produced in a layered process and are not easily interpretable. Therefore, the use of simple classifiers cannot provide high efficiency. Therefore, our suggestion is to use a tunable classifier, such as MLP. As can be seen, none of the classifiers that were compared provide higher accuracy than the proposed MLP. All the models that were compared in Table 4 have been implemented by us. In all experiments, we used the same dataset, same validation technique, and performance evaluation metric to have a fair comparison. K-nearest neighbors is performed based on the Euclidean metric as a distance criterion and three different K values (K = 3, 5, & 7). Random forest is performed based on two different numbers of trees (N = 20, 50). Different max. depth values (M = 1, 3, 5 & 10) were tested. In both of them, M = 3 provides the highest accuracy, which is reported in Table 4. Naïve Bayes is performed as an efficient implementation of the Bayes theorem. The prior probability for each class is calculated from the training data and considered independent of other classes (conditionally independent). The J-48 tree is evaluated based on different confidence factors and different minimum numbers of instance per leaf. Finally, confidence factor = 0.25 and the minimum number of instance per leaf = 3 provide maximum accuracy.

### 4.5. Comparison with State-of-the-Art Methods

The performance of the proposed method is compared with some efficient approaches in this scope in terms of accuracy, as reported in Table 6 and Table 7. In this respect, to have a fair comparison, the same validation conditions (K-folds) and same dataset (2d-Hela) are considered in experiments. To have a fair comparison, all compared results in Table 5 are compared with precision based on the reported results in related references. Some papers did not report the performance of the proposed approach in two types, two-class and seven-class problems. So, the phrase “NR” in Table 6 and Table 7 means not reported. The methods entitled “baseline” in Table 5, have been implemented by us. In both basic networks (ResNet-34 and Vgg-19) that have been compared, the way to set the hyper-parameters, number of train images, database and the evaluation metrics have been the same with the proposed method. In both baseline networks, a fully connected layer with two/seven neurons (equal to the number of classes) is used as the final layer for classification.

As shown in Table 6, the proposed method provides higher classification accuracy than all compared methods in the two-class classification manner. Additionally, the results demonstrate that our proposed approach has a better performance than nearly all of the compared methods in the seven-class classification problem. ConvNet-T [11] provides the highest accuracy, which is about 0.16 percent higher than our proposed method in the 7-class problem. The accuracy of ConvNet-T [11] is not reported in the two-class classification problem in [11]. The structure of ConvNet-T is designed for Pap smear classification and it has been trained using related Pap smear images. However, our proposed approach uses two popular deep networks, which can be used in the pre-trained format too. So, it can be used in other computer vision applications, such as medical image classification.

### 4.6. Computational Complexity Evaluation

In this article, deep neural networks are used just for feature extraction. Therefore, some of the final layers in these networks, which are related to the classification activity, have been removed. As mentioned in the introduction, the computational complexity and runtime of the proposed method are somewhat lower than deep neural networks. In Table 8, the runtime required to classify 20 random input images and basic deep networks in the presented method has been evaluated, and as can be seen, removing the fully connected or softmax layers reduced the total runtime.

### 4.7. Contribution Justification

The main contribution of this paper was that using the main strategy of machine learning algorithms to separately implement the feature extraction and classification phases while deep features that are extracted from CNNs are used in feature extraction phases, may improve the performance of cervical cancer diagnosis. The reported results in Table 2 show that adjusting the number of hidden neurons has an effect on the performance of cervical cancer diagnosis. Additionally, our proposed methods of choosing hidden neurons provide higher accuracy than random selection theory.

Reported results in Table 4 and Table 5 prove that the tuned MLP provides higher accuracy than simple linear/non-linear supervised classifiers. The reported results in Table 6 and Table 7 prove that using MLP fed with deep features provides higher diagnosis rates than using base-line deep networks for classification.

Additionally, the results in Table 8, proves that using MLP instead of classification layers of deep networks, decreases the runtime of the classification process on average.

## 5. Conclusions

The main goal of this paper was to diagnose cervical cancer with acceptable accuracy. According to recent studies, deep networks provide higher accuracy than handcrafted features. In this regard, the combination of two deep networks (ResNet34 and VGG-19) and a classic multi-layer neural network was proposed in this article. To increase the classification accuracy, the layers related to the classification step of both deep networks were removed and replaced with a flatten layer. Then, a multi-layer neural network was fed based on the extracted deep features and performed the classification process. The results showed that the presented method provides higher accuracy than using each of the two networks separately. Additionally, the comparative results showed that the presented method provides higher accuracy than many existing methods. The computational complexity of the presented method is higher than some deep networks, but due to the quality of the hardware in the test phase, it can provide the result to the expert in an acceptable time. The method presented in this article can be extended to address some problems of visual pattern classification. Therefore, as a suggestion for future studies, the method presented in this article can be used in other fields of computer vision such as image retrieval, skin cancer detection, etc. As discussed in Section 1.2, the main contribution of this paper was to propose an approach for cervical cancer diagnosis based on a machine learning strategy that is fed with deep features. So, different deep networks (such as ResNet versions, GoogleNet, MobileNet, etc.) can be used to extract deep features. This paper presents a general approach of using the potential of deep features in a machine-learning-based structure. The proposed tuned MLP can be fed with different deep features. One of the advantages of the method presented in this article is that it can be generalized. So, using different deep networks, such as Google Net, is suggested as for future works.

## Figures and Tables

**Figure 1 diagnostics-13-00686-f001:**
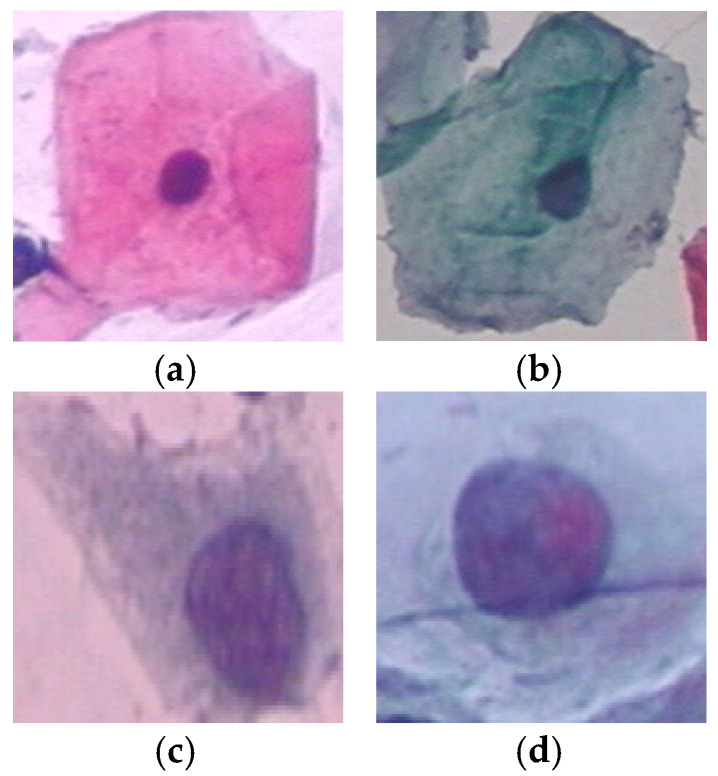
Some samples of cervix cells from the Herlev dataset [3]. (**a**) Normal cell (normal superficial class); (**b**) Normal cell (intermediate class); (**c**) Abnormal cell (severe dysplastic class); (**d**) Abnormal cell (moderate dysplastic class).

**Figure 2 diagnostics-13-00686-f002:**
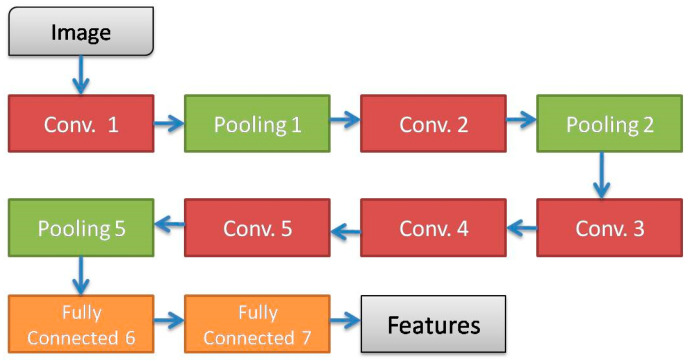
The proposed CNN in [10].

**Figure 3 diagnostics-13-00686-f003:**
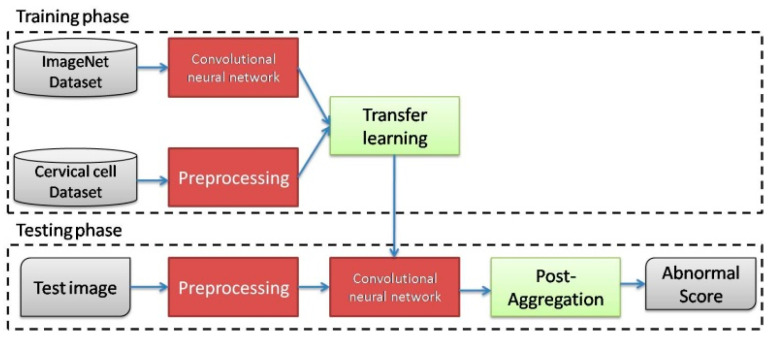
The total structure of the cervical cancer detection method in [11].

**Figure 4 diagnostics-13-00686-f004:**
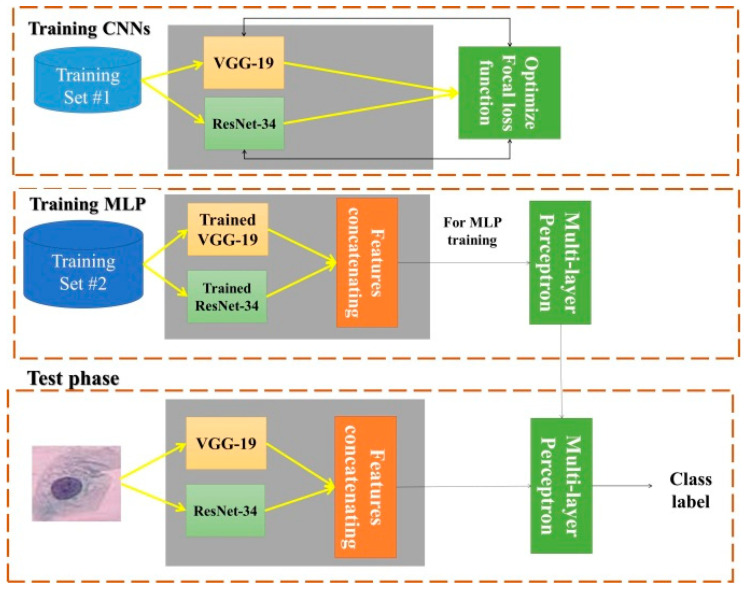
The main block diagram of proposed cervical cancer diagnosis method.

**Figure 5 diagnostics-13-00686-f005:**
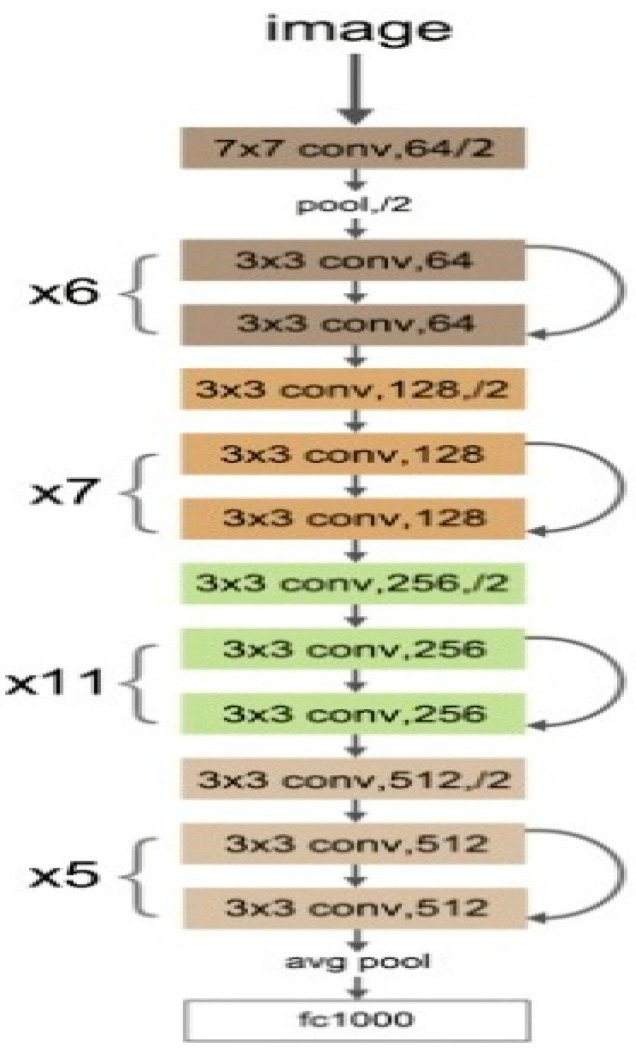
The main architecture of deep ResNet-34 network.

**Figure 6 diagnostics-13-00686-f006:**
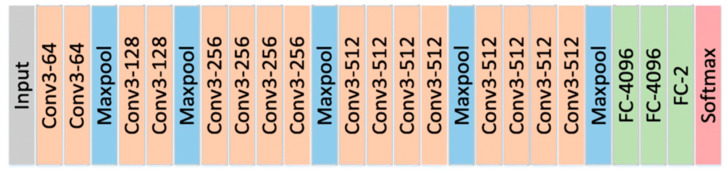
The main architecture of VGG-19 network.

**Figure 7 diagnostics-13-00686-f007:**
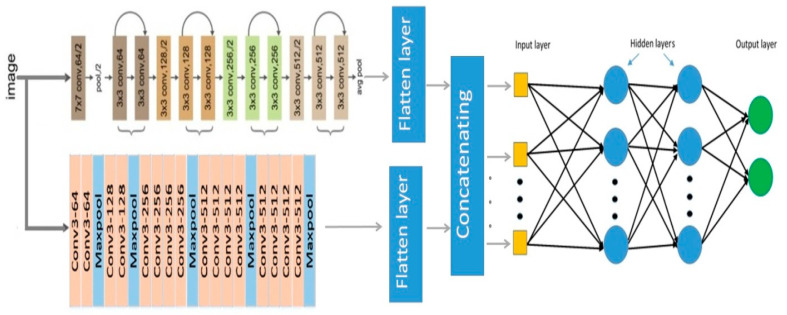
The main structure of proposed multi-layer perceptron fed with VGG-19 and ResNet-34.

**Figure 8 diagnostics-13-00686-f008:**
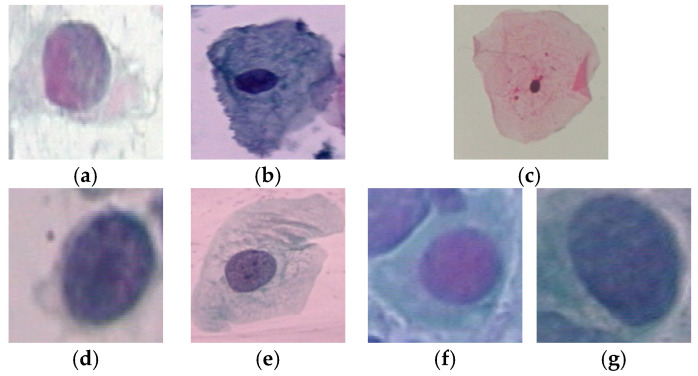
Some samples of the Herlev dataset [3]. (**a**) Normal columnar cell; (**b**) Normal intermediate cell; (**c**) Normal superficial cell; (**d**) Abnormal carcinoma in situ cell; (**e**) Abnormal light dysplastic cell; (**f**) Abnormal cell moderate dysplastic cell; (**g**) Abnormal severe dysplastic cell.

**Table 1 diagnostics-13-00686-t001:** The performance evaluation based on different learning rates in CNNs training process in terms of accuracy (%).

Learning Rate (First 10 Epochs)	Learning Rate (Second 20 Epochs)	Weight Decay Value	Accuracy
10^−3^	10^−4^	10^−3^	97.31
10^−3^	10^−4^	10^−2^	97.47
10^−3^	10^−5^	10^−3^	98.32
10^−3^	10^−5^	10^−2^	98.45
10^−4^	10^−5^	10^−3^	98.88
10^−4^	10^−5^	10^−2^	98.95
10^−5^	10^−6^	10^−3^	99.24
10^−5^	10^−6^	10^−2^	99.63
10^−5^	10^−5^	10^−3^	98.97
10^−5^	10^−5^	10^−2^	99.16

**Table 2 diagnostics-13-00686-t002:** The performance evaluation of the proposed approach based on different variables of MLP in terms of accuracy, precision, and recall (%).

	Problem	2 Classes			7 Classes		
Number of Hidden Neurons		Accuracy	Precision	Recall	Accuracy	Precision	Recall
10		96.95	97.11	97.24	96.53	96.69	96.84
20		97.26	97.42	97.57	97.32	97.48	97.65
30		98.73	98.89	99.04	97.41	97.57	97.72
40		98.54	98.70	98.85	96.91	97.08	97.20
Number of classes		98.95	99.11	99.26	97.49	97.66	97.80
Number of attributes		98.52	98.68	98.82	96.34	96.52	96.65
Number of attributes + Number of classes		98.89	99.05	99.21	96.89	97.06	97.21
(Number of attributes + Number of classes)/2		99.23	99.40	99.55	97.60	97.77	97.90

**Table 3 diagnostics-13-00686-t003:** The confusion matrix of the proposed cervical cancer classification on the Herlev dataset [3].

	Normal	Abnormal
Normal	238	4
Abnormal	3	672

**Table 4 diagnostics-13-00686-t004:** The performance evaluation of proposed approach based on different classifiers on the Herlev dataset in terms of accuracy (%).

	Problem	2 Classes	7 Classes
Classifier			
KNN, K = 3		96.02	94.43
KNN, K = 5		98.19	96.61
KNN, K = 7		97.14	95.52
Naïve Bayes		92.83	91.34
Random forest (Number of trees = 20)		97.14	95.51
Random forest (Number of trees = 50)		97.01	95.48
J-48 Tree		91.49	90.06
Proposed MLP (Fed with ResNet-34 + Vgg-19)		99.23	97.65
Proposed MLP (Fed with ResNet-50 + Vgg-19)		99.32	97.65

**Table 5 diagnostics-13-00686-t005:** The performance evaluation of proposed approach based on different classifiers on the SIPAKMED dataset in terms of accuracy (%).

	Problem	5 Classes
Classifier		
KNN, K = 3		97.39
KNN, K = 5		99.59
KNN, K = 7		98.49
Naïve Bayes		94.26
Random forest (Number of trees = 20)		98.47
Random forest (Number of trees = 50)		98.50
J-48 Tree		92.88
Proposed MLP (Fed with ResNet-34 + Vgg-19)		99.64
Proposed MLP (Fed with ResNet-50 + Vgg-19)		99.71

**Table 6 diagnostics-13-00686-t006:** Comparison results on the Herlev dataset in terms of accuracy (%).

	Problem	2 Classes	7 Classes
Approach			
GLCM + LBP + DWT + ANN [6]		NR	88.2
GSV + SF + IQP [7]		88.47	NR
AlexNet + Feature selection [10]		89.97	NR
ConvNet-T [11]		NR	97.7
{Shape + Histogram} + Ensemble [12]		96.5	NR
{Shape + Histogram} + MLP [12]		98.88	NR
MT-ULTP + Optimized MLP [15]		98.90	96.63
PSO + 1NN [35]		NR	96.7
Texture + Random forest [36]		98.42	94.76
Texture + Ridge [36]		95.76	92.82
Random forest + Hierarchical [36]		NR	95.43
Ridge + Hierarchical [36]		NR	94.34
GoogleNet [37]		94.51	NR
Graph cut + MLP [38]		94.3	NR
DeepCell-V2 [18]		NR	92.71
ResNet-34 (base line network)		95.02	93.22
ResNet-50 (base line network)		95.38	93.27
VGG-19 (base line network)		93.87	92.76
Proposed method (ResNet-34 + Vgg-19 + Tuned MLP)		99.23	97.65
Proposed method (ResNet-50 + Vgg-19 + Tuned MLP)		99.32	97.65

**Table 7 diagnostics-13-00686-t007:** Comparison results on the SIPKAMED dataset in terms of accuracy (%).

	Problem	5 Classes
Approach		
DeepCell-V1 [18]		95.25
DeepCell-V2 [18]		95.62
DeepCyto + RF [39]		96.35
DeepCyto + ANN [39]		99.71
ResNet-34 (base line network)		95.12
ResNet-50 (base line network)		95.47
VGG-19 (base line network)		94.01
Proposed method (ResNet-34 + Vgg-19 + Tuned MLP)		99.64
Proposed method (ResNet-50 + Vgg-19 + Tuned MLP)		99.71

**Table 8 diagnostics-13-00686-t008:** Runtime evaluation of the proposed approach and baseline networks for 10 random selected image classification.

Approach	Runtime per One Image Input (ms)
ResNet-34 (base line network)	970
ResNet-50 (base line network)	5541
VGG-19 (base line network)	2950
Proposed method (ResNet-34 + VGG-19 + Tuned MLP)	930
Proposed method (ResNet-50 + VGG-19 + Tuned MLP)	5210

## Data Availability

The datasets generated during and/or analyzed during the current study are available from the corresponding author on reasonable request.

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
