# Peer review of "Developing a Tuned Three-Layer Perceptron Fed with Trained Deep Convolutional Neural Networks for Cervical Cancer Diagnosis"

_diagnostics, 2023, doi:10.3390/diagnostics13040686_

Round 1

Reviewer 1 Report

This paper proposes a method to detect cervical cancer based on cervical smear images. The method is divided into feature extraction stage and feature classification stage. The method of CNN is used in feature extraction stage and the method of MLP is used in feature classification stage. This work is very effective for the early diagnosis and treatment of cervical cancer, but there are some problems as follows:

1. The proposed method is more like some integration of previous work, which is not novel enough.

2. The line 72 in the article begins to introduce the shortcomings of the high computational complexity of CNN method, and the line 76 begins to introduce that the proposed method has some optimization in computational complexity, but the feature extractor of the proposed method is still two convolutional neural networks. Can you add some evaluation indicators of computational complexity to show the advantages of the proposed method at this point?

3. In order to extract better features, will it be better to replace ResNet34 with ResNet50? VGG19 also has this problem.

4. How about replacing ResNet34 with Google Net? It may be better to add a group of ablation experiments on ResNet34 and VGG19.

5. What theory do the four methods listen to for selecting the number of MLP hidden layers? Change it to "Number of attributes × 2 "Is it OK?

6. In order to prove the function of the proposed module "Tuned MLP", do you need a set of pure convolutional neural network models integrating ResNet34 and VGG19 for comparison?

Author Response

Authors Response to the Reviewer Comments

Journal:    Diagnostics          

Manuscript ID:    diagnostics-2150924 (Revised version)

Title of Paper:    Developing a tuned three-layer perceptron fed with trained deep convolutional neural networks for cervical cancer diagnosis

Authors:  Shervan Fekri-Ershad, Marwa Fadhil Alsaffar

We appreciate the time and efforts by the editor and referees in reviewing this manuscript. The authors would like to thank reviewers for providing us with their comments and suggestion to improve the quality of the manuscript. We have addressed all issues indicated in the review report in a point–by-point fashion, and believed that the revised version can meet the journal publication requirements.

Response to Comments from Reviewer 1

Comment #

Comments and Corrections

Address

1

Comment:

The proposed method is more like some integration of previous work, which is not novel enough.

Corrections:

Thank you for your attentions.

The main goal of this article is to present a method for diagnosing cervical cancer based on Pap smear image analysis.Most of the methods that have been presented so far can be classified into two categories, based on machine learning or deep learning-based. Feature engineering is not used in deep learning-based methods, and feature extraction and classification steps are not separate.

Therefore, As explained in the revised sub-section 1-2, the main contribution of this paper is to present a method based on machine learning strategy for cervical cancer diagnosis which use deep features.

Therefore, in this article, the feature extraction stage is done separately based on deep networks, and a multi-layer perceptron network is used in the classification stage.

Different deep networks (such as ResNet, VGG, GoogleNet, etc) can be used to extract deep features. But this article is not a combination of several deep networks. This paper presents a general approach for using the potential of deep features in a machine learning-based structure. In the classification stage, a MLP is used, with some simple ideas to tune the number of hidden neurons to increase the detection accuracy. More mathematical details of how to define the perceptron neural network and its relationship with the output of deep features were presented in the revised version. We hope that the added technical details will make it more clear that the main contribution of this method is not to combine previous works. One of the main advantages of the method presented in this article is that it can be generalized. Our research shows that so far no paper has used the deep features extracted from two deep networks, VGG-19 and ResNet-34, together with the tuned three-layer perceptron neural network, to diagnose cervical cancer types.Based on your suggestion, the main contribution and novelties is described with more details and more clear in the revised version.

Page 3, Section 1.2

2

Comment:

The line 72 in the article begins to introduce the shortcomings of the high computational complexity of CNN method, and the line 76 begins to introduce that the proposed method has some optimization in computational complexity, but the feature extractor of the proposed method is still two convolutional neural networks. Can you add some evaluation indicators of computational complexity to show the advantages of the proposed method at this point?

Corrections:

Thank you for bringing this efficient comment to our attention.

In this article, deep neural networks are used just for feature extraction. Therefore, some of the final layers in these networks, which are related to the classification activity, have been removed. As mentioned in the introduction, the computational complexity and runtime of the proposed method is somewhat lower than deep neural networks because of removing these layers.

Based on your suggestion, the runtime of the proposed method is evaluated in sub-section 4.7 separately. In the Table 8, the run time required to classify 20 random input image in the presented method and basic deep networks has been evaluated, and as can be seen, removing the fully connected or softmax layers, reduced the total runtime.

Pages 15 & 16

Section 4.6

3

Comment:

In order to extract better features, will it be better to replace ResNet34 with ResNet50? VGG19 also has this problem.

How about replacing ResNet34 with Google Net? It may be better to add a group of ablation experiments on ResNet34 and VGG19.

Corrections:

As explained in the revised sub-section 1-2, the main contribution of this paper is to present a method based on machine learning strategy and deep features, to detect cervical cancer types. Therefore, in this article, the feature extraction stage is done separately based on deep networks, and a multi-layer perceptron network is used in the classification stage.

We agree with you that different deep networks (such as ResNet-50, GoogleNet, etc) can be used to extract deep features. But this article is not a combination of several deep networks. This paper presents a general approach for using the potential of deep features in a machine learning-based structure. In the classification stage, a MLP is used, with some simple ideas to tune the number of hidden neurons to increase the detection accuracy. More mathematical details of how to define the perceptron neural network and its relationship with the output of deep features were presented in the revised version. We hope that the added technical details will make it more clear that the main contribution of this method is not to combine deep multi-networks.We agree with you that different deep networks can be used to extract deep features. One of the advantages of the method presented in this article is that it can be generalized. Based on your suggestion, we also evaluated the performance of the proposed method based on ResNet-50 instead of ResNet-34. and the results were reported in Tables 4-7. Also, the runtime of the proposed algorithm was evaluated based on ResNet-50 and presented in the Table 8. As can be seen in the Table 8, the runtime of proposed method using ResNet-50 is higher than ResNet-34.Based on your suggestions in this comment and some previous comments, the main contribution is described with more details in a clear way in the revised version. Also, potential to use other deep neural networks is discussed as future work in the conclusion. 

Page 9, Section 3.3

Tables 4-8

4

Comment:

What theory do the four methods listen to for selecting the number of MLP hidden layers? Change it to "Number of attributes × 2 "Is it OK?

Corrections:

We agree with you, that we have not proposed a straightforward algorithm to choose number of hidden neurons. We don’t claim any straightforward algorithm to choose hidden neurons in the text. We do not claim that the proposed 4 theories give the best results in all possible cases.

These 4 theories are proposed based on [31]. We only claim that these 4 theories are applicable to most classification problems and, as shown in the results, provide higher accuracy than the compared methods in this domain. Also, We agree with you that the theories should be explained in more detail.  

Based on your comment, following explanations regarding the theory have been added to the text.

The number of neurons in the hidden layer of the MLP network is an adjustable parameter that also plays role in the final classification accuracy. Because in this article, deep networks are used for feature extraction, usually the dimensions of the feature map in the pre-end layers in these networks are large. Logically number of neurons in the MLP network can be any integer value, so determining a logical limitation for it that makes it applicable in all other problems plays important role in choosing the 4 proposed ideas.

Therefore, we suggested 4 ideas based on following issue:

1) The number of features and the number of classes are the only parameters common to all classification problems. Therefore, these two parameters were used in all 4 proposed ideas.

2) In most classification problems, the number of features is usually more than the number of classes. In some databases, the number of classes is slightly more than the number of features. Therefore, it was tried to limit the number of neurons in the 4 proposed ideas based on these two points and their average.

It is possible to choose "Number of attributes × 2" as an idea. We evaluated the performance of the proposed approach based on your suggested idea. Results show that this idea provides 98.73 percent accuracy in 2-class problem and   96.81 percent accuracy on 7-class problem.

Pages 9 & 10

Section 3.3.1

5

Comment:

In order to prove the function of the proposed module "Tuned MLP", do you need a set of pure convolutional neural network models integrating ResNet34 and VGG19 for comparison?

Corrections:

As explained in comment 5, we do not claim to provide a step-by-step algorithm for determining the number of hidden neurons. We have tried to increase the final accuracy of the cervical cancer detection method by tuning the number of hidden neurons. Therefore, we have used these 4 ideas based on [31], in this problem. It has been shown in the results that tuned MLP module provides higher accuracy than the existing methods.

Pages 9 & 10

Section 3.3.1

Reviewer 2 Report

The paper deals with an interesting topic from the medical domain. The paper should not be accepted for publication in its present form. I mention some recommendations for the authors.

- Novelties are superficially presented in this paper. The claims should seriously support each novelty. Especially rigorous experimental evaluation
must confirm that.

- Images are of terrible resolution. Hence, provide images with a higher aspect ratio. Use only your figures.

- I would like to see also more mathematical approach to presenting the algorithms. Otherwise, this method is just a mix/hybrid of several ways that are connected.

- You must show all the parameter settings of all methods. Please give us a piece of evidence that you ensured fair comparisons.

- More datasets should be included in this paper.

- Give a more detailed view of the practical use of this method.

Author Response

Authors Response to the Reviewer Comments

Journal:    Diagnostics          

Manuscript ID:    diagnostics-2150924 (Revised version)

Title of Paper:    Developing a tuned three-layer perceptron fed with trained deep convolutional neural networks for cervical cancer diagnosis

Authors:  Shervan Fekri-Ershad, Marwa Fadhil Alsaffar

We appreciate the time and efforts by the editor and referees in reviewing this manuscript. The authors would like to thank reviewers for providing us with their comments and suggestion to improve the quality of the manuscript. We have addressed all issues indicated in the review report in a point–by-point fashion, and believed that the revised version can meet the journal publication requirements.

Response to Comments from Reviewer 2

Comment #

Comments and Corrections

Address

1

Comment:

Novelties are superficially presented in this paper. The claims should seriously support each novelty. Especially rigorous experimental evaluation
must confirm that.

Corrections:

Thank you for bringing this efficient comment to our attention.

The main goal of this article is to present a method for diagnosing cervical cancer based on Pap smear image analysis.Most of the methods that have been presented so far can be classified into two categories, based on machine learning and deep learning based. Feature engineering is not used in deep learning-based methods, and feature extraction and classification steps are not separate.

·         Therefore, As explained in the revised sub-section 1-2, the main contribution of this paper is to present a method based on machine learning strategy for cervical cancer diagnosis which use deep features instead of feature engineering.

Deep features are produced in a layered process and are not easily interpretable. Therefore, the use of simple classifiers cannot provide high efficiency. Therefore, our suggestion is to use a tunable classifier such as MLP. The second contribution of this paper is as follows:

·         Tuning the number of hidden neurons in the MLP which fed with deep features to increase the performance of the cervical cancer diagnosis.

The reported results in the Table 2, shows that tuning the number of hidden neurons effect on the performance of cervical cancer diagnosis. Also, our proposed methods to choose hidden neurons provide higher accuracy than randomly selection theory.

Reported results in the Table 4 & 5 shows that adjusted MLP provide higher accuracy than simple linear/non-linear supervised classifiers.

Reported results in the Tables 6 & 7, proves that using MLP fed with deep features, provide higher diagnosis rate than using base-line deep networks for classification. 

Also, added results in the Table 8, proves that using MLP instead of classification layers of deep networks, decrease runtime of classification process.

Based on your suggestion, contribution justification is discussed with more details in the section 4.7.

Page 16, Section 4.7

2

Comment:

Images are of terrible resolution. Hence, provide images with a higher aspect ratio.

Corrections:

Based on your suggestion, images from dataset with higher resolution were added to the text. 

Whole images

3

Comment:

I would like to see also more mathematical approach to presenting the algorithms. Otherwise, this method is just a mix/hybrid of several ways that are connected.

Corrections:

Thank you for bringing this efficient comment to our attention. Based on your comment, mathematical details about the MLP learning process, How to feed neural network from extracted deep features, weights initialization and hyper-parameter optimization is added to the text in the revised version. 

Pages 9-11

Section 3.3.1

&

section 3.3.2

4

Comment:

You must show all the parameter settings of all methods. Please give us a piece of evidence that you ensured fair comparisons.

Corrections:

Thank you for bringing this efficient comment to our attention.

All the models that were compared in the Tables 4 & 5 have been implemented by us. In all experiments, we used same dataset, same validation technique and performance evaluation metric to have a fair comparison. Based on your suggestion, more details about compared methods in the Table 4, such as parameter settings are described in the revised version.

All the results presented in the Table 6 & 7, in relation to compared methods, are exactly written based on the reported results in related references. In order to make the comparison fair, in all of the chosen methods for comparison, database, validation technique and the evaluation criteria have been the same.

The methods entitled “baseline” in the table 6, have been implemented by us.In basic networks (ResNet-34, ResNet-50 and Vgg-19), that have been compared, the way to set the hyper-parameters, number of train images, database and the evaluation metrics have been the same with the proposed method.

Based on your suggestion, more details regarding the parameter settings of the compared methods in both Tables 4-6 were added to the text.

Page 14

Section 4.4 &

section 4.5

5

Comment:

More datasets should be included in this paper.

Corrections:

Thank you for your efficient comment. Based on your suggestion, more than Herlev dataset, the performance of the proposed approach is evaluated on SipkaMED dataset in the revised version too. Experimental results based on different classifiers are reported in the Table 5. Also, it is compared with some efficient methods in this scope with a same dataset and validation technique, which is reported in the Table 7. The SipkaMed dataset details are described in the section 4.1.  

Page 12, section 4.1

Tables 5 & 7

6

Comment:

Give a more detailed view of the practical use of this method.

Corrections:

The main purpose of this article is to present a method for diagnosing cervical cancer based on the analysis of Pap smear images. Every year, hundreds of women around the world die from cervical cancer. Early and accurate diagnosis can increase the patient's chance of survival. So far, many articles have been presented in this field. In some areas of the world where it is difficult to access a specialist doctor, the automatic diagnosis system can help early diagnosis of patients and connect definitive patients to the nearest specialist doctor and surgeon. Also, costs for patients and insurance companies will decrease. Automatic detection systems can be replicated in all regions around the world thanks to simple software.Based on your suggestion, above discussion about the practical use of the automatic diagnosis systems is added to the text.

Page 2, Section 1

Round 2

Reviewer 1 Report

The paper can be accepted in present form.